# REASONING CURRICULUM: BOOTSTRAPPING BROAD LLM REASONING FROM MATH

## ABSTRACT

Reinforcement learning (RL) can elicit strong reasoning in large language models (LLMs), yet most open efforts focus on math and code. We propose `Reasoning Curriculum`, a simple two-stage curriculum that first elicits reasoning skills in pretraining-aligned domains such as math, then adapts and refines these skills across other domains via joint RL. Stage 1 performs a brief cold start and then math-only RL with verifiable rewards to develop reasoning skills. Stage 2 runs joint RL on mixed-domain data to transfer and consolidate these skills. The curriculum is minimal and backbone-agnostic, requiring no specialized reward models beyond standard verifiability checks. Evaluated on Qwen3-4B and Llama-3.1-8B over a multi-domain suite, `Reasoning Curriculum` yields consistent gains. Ablations and a cognitive-skill analysis indicate that both stages are necessary and that math-first elicitation increases cognitive behaviors important for solving complex problems. `Reasoning Curriculum` provides a compact, easy-to-adopt recipe for general reasoning.

## 1 INTRODUCTION

Recent work has advanced rapidly on eliciting reasoning in large language models (LLMs). Chain-of-Thought (CoT) prompting (Wei et al., 2022) asks models to produce intermediate steps before answering and substantially improves reasoning performance. Building on this idea, proprietary systems train with reinforcement learning (RL) to refine long chains of thought, achieving strong results in competition math and programming (OpenAI, 2024). Open-source efforts follow a similar trajectory, reporting competitive performance and exposing training practices to broader scrutiny (Team, 2024; Guo et al., 2025; Zeng et al., 2025; Luo et al., 2025b;a).

Despite this progress, most open-source works concentrates on math and code, domains with abundant data and easily verifiable rewards. General reasoning across diverse domains remains comparatively underexplored. Recent work expands beyond math and code (Akter et al., 2025; Ma et al., 2025; Cheng et al., 2025) and focuses on curating data across broad domains, yet effective, cross-domain training strategies for strong reasoning models are still scarce.

We start from a premise suggested by the literature and our preliminary experiments: math is unusually amenable to RL-based skill elicitation. Significant gains can arise even under weak supervision, including spurious or random rewards, and sometimes from very small training sets (Shao et al., 2025b; Wang et al., 2025). We hypothesize that math serves as an effective driver for discovering core reasoning skills that can later be adapted to other domains through on-policy training.

This paper proposes **Reasoning Curriculum**, a simple two-stage curriculum. Stage 1 elicits reasoning via supervised cold start and math-only RL. Stage 2 transfers and refines the learned skills by running joint RL on a mixed-domain corpus spanning math, STEM, code, simulation, logic, and tabular tasks. The curriculum is intentionally minimal, requires no specialized reward models beyond standard verifiability checks, and applies across backbones.

We evaluate `Reasoning Curriculum` on Qwen3-4B and Llama-3.1-8B. On Qwen, our 4B model consistently outperforms similarly sized baselines and is competitive with, and sometimes exceeds, 32B systems. On Llama, directly porting the Qwen recipe yields only small gains, so we introduce a simple difficulty curriculum within the Math-RL stage (medium then hard). With this change, the curriculum again improves performance across all domains.

We also provide evidence for the mechanism behind `Reasoning Curriculum` through ablations and a cognitive-skill analysis, showing that math-first elicitation increases transferable behaviors and that both stages are necessary for the full gains.

In summary, `Reasoning Curriculum` follows a simple strategy: first develop reasoning skills in pretraining-aligned domains such as math using verifiable rewards, then adapt and refine them across diverse domains with joint RL. This yields a compact, easy to adopt training recipe for general reasoning that consistently improves performance across domains.

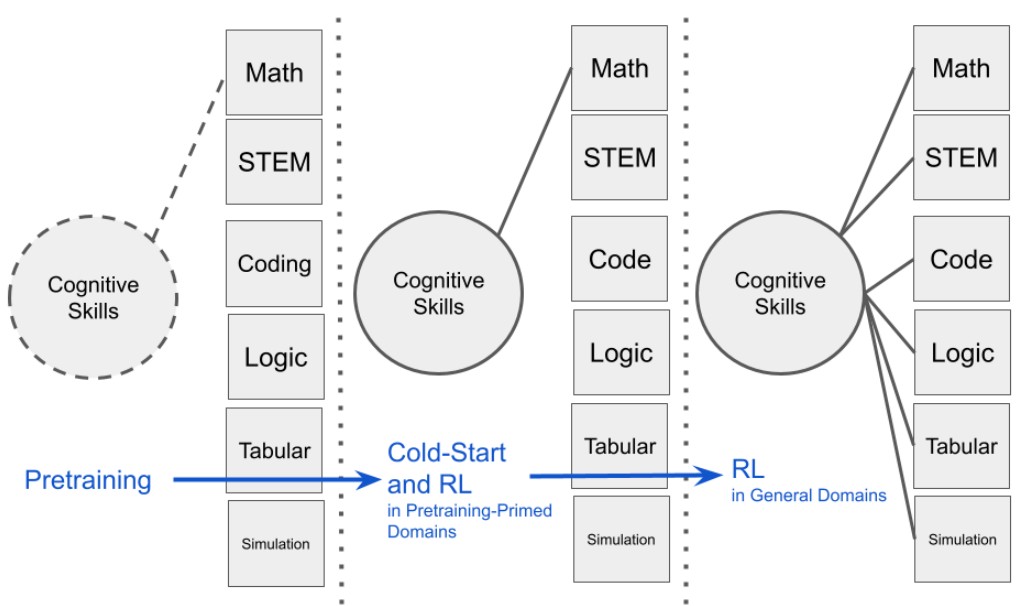

Figure 1: Reasoning curriculum overview. Stage 0 (pretraining, not conducted in this work): cognitive skills exist but are weakly expressed on data-rich domains like math. Stage 1 (cold-start + math-only RL): skills are elicited and strengthened in pretraining-primed domains. Stage 2 (joint RL): skills are transferred and refined across general domains (code, logic, tabular, simulation). Blue arrows indicate the training progression.

## 2 REASONING CURRICULUM

Suppose $(x, y)$ is a question–answer pair and $z$ is a chain of thought that produces $y$. The reasoning process often manifests distinct cognitive skills. Four skills, commonly observed in both human solvers and successful LLMs (Gandhi et al., 2025; Zeng et al., 2025), are:

- Subgoal setting: decomposing a complex problem into smaller, manageable steps.
- Enumeration: considering multiple cases or possibilities.
- Backtracking: identifying errors during generation and explicitly revising prior steps.
- Verification: checking intermediate results to ensure correctness.

While subgoal setting and enumeration frequently appear in most modern LLMs with CoTs, verification and backtracking are often associated with LongCoT models such as Deepseek-R1 (DeepSeek-AI et al., 2025) and are critical for solving harder problems. Our goal is to increase the use of these skills in general domains and thereby strengthen LLM reasoning.

It is frequently observed that reinforcement learning with verifiable rewards (RLVR) on math data increases the use of these skills and yields substantial gains (Zeng et al., 2025; Luo et al., 2025b;a; Hu et al., 2025b), even under noisy rewards (Shao et al., 2025a). Given the readiness of skill elicitation in the math domain, we hypothesize that pretraining already exposes models to these skills in data-rich domains such as math, making them easier to elicit during post-training.

We therefore propose a two-stage reasoning curriculum (Figure 1). First, we elicit skills on math via a brief cold start followed by reinforcement learning with verifiable rewards. Second, we refine and adapt these skills through joint RL on mixed-domain data to improve general reasoning.

## 2.1 MATH TRAINING

### 2.1.1 COLD START

Given a pretrained LLM, we first perform supervised fine-tuning on a small set of math examples to expose the model to skill-rich thought traces:

$$\mathcal{J}_{\text{Cold-Start}}(\theta) = \mathbb{E}_{(x,z,y)\sim\mathcal{D}_{\text{CS}}}[\log \pi_\theta(y, z \mid x)]. \tag{1}$$

Although recent work explores a zero-RL setup that applies RL without any supervised LongCoT training (Hu et al., 2025a; Zeng et al., 2025), in practice strong reasoning systems almost always begin with some cold-start supervision. Even within the DeepSeek-R1 line, which popularized the zero-RL idea, widely used variants include supervised components (DeepSeek-AI et al., 2025). We therefore adopt a brief cold start. It quickly exposes the model to diverse reasoning skills and creates a realistic setting to study how SFT interacts with RL. Empirically, cold start helps the model imitate multiple cognitive skills, while on-policy RL is still critical to consolidate these behaviors into measurable gains in reasoning performance (see Section 4.3 for detailed discussions).

### 2.1.2 MATH RL

For RL, Group Relative Policy Optimization (GRPO) (Shao et al., 2024) has become popular due to its efficiency and the success of DeepSeek-R1 (DeepSeek-AI et al., 2025). We use the DAPO variant (Yu et al., 2025), which introduces several modifications that improve stability and performance:

$$\mathcal{J}_{\text{DAPO}}(\theta) = \mathbb{E}_{(x,y)\sim\mathcal{D},\{y_i\}_{i=1}^G\sim\pi_{\theta_{\text{old}}}(\cdot|x)}$$

$$\left[\frac{1}{\sum_{i=1}^G |y_i|}\sum_{i=1}^G\sum_{t=1}^{|y_i|}\min\left(r_{i,t}(\theta)\hat{A}_{i,t},\ \text{clip}\left(r_{i,t}(\theta), 1-\varepsilon_{\text{low}}, 1+\varepsilon_{\text{high}}\right)\hat{A}_{i,t}\right)\right] \tag{2}$$

$$\text{s.t.}\quad 0 < \left|\{y_i \mid \texttt{is\_equivalent}(y, y_i)\}\right| < G,$$

where

$$r_{i,t}(\theta) = \frac{\pi_\theta(y_{i,t} \mid x, y_{i,<t})}{\pi_{\theta_{\text{old}}}(y_{i,t} \mid x, y_{i,<t})}, \quad \hat{A}_{i,t} = \frac{R_i - \text{mean}(\{R_i\}_{i=1}^G)}{\text{std}(\{R_i\}_{i=1}^G)}. \tag{3}$$

The constraint filters groups so that at least one sample is correct and at least one is incorrect, which makes relative advantages meaningful. Also, we omit the KL penalty to encourage exploration.

Following Zeng et al. (2025), we avoid format rewards that can hinder exploration and use only correctness as the outcome reward:

$$R(\hat{y}, y) = \begin{cases} 1, & \texttt{is\_equivalent}(\hat{y}, y) \\ 0, & \text{otherwise.} \end{cases} \tag{4}$$

## 2.2 JOINT RL

After the Math-focused stage, we train a single policy with joint RL across our full suite of domains (Math, Code, STEM, Logic, Simulation, Tabular; see Experiments for details). Training uses the same DAPO objective as in Equation 2; only the reward computation differs by domain. Unless noted otherwise, rewards are binary $R \in \{0,1\}$ (1 if the prediction matches the ground truth, 0 otherwise). Two Logic datasets permit partial credit, so we assign $R \in (0,1)$ when appropriate (see Experiments 3.1). All rewards are derived automatically from verifiable signals and are therefore low noise, which is the key to stable and effective RL. Following prior work on general reasoning (Ma et al., 2025; Cheng et al., 2025), we combine three evaluation strategies to accommodate domain-specific answer formats:

- Rule-based matching. Used in Math, Logic, Simulation, and Tabular. The model is prompted to place the final answer in a prescribed format (e.g., \boxed{}). We extract and normalize the answer, then compare it with the ground-truth for exact or numeric equivalence.

- Model-based equivalence. Used in STEM where questions have free-form answers and deterministic rules are brittle. An LLM is used to compare the model output with the reference answer for semantic equivalence. This method robustly handles phrasing differences while maintaining low reward noise.
- Execution-based verification. Used in Code. The generated function or script is executed against a unit-test suite and receives a reward of 1 only if all tests pass, and 0 otherwise.

## 3 EXPERIMENTS

### 3.1 TRAINING DATA

**Cold Start Data** We randomly sample 20k problems from NuminaMath (Li et al., 2024) and generate responses with DeepSeek-R1 (DeepSeek-AI et al., 2025). We retain 10k examples whose R1 responses produce correct answers and use them for cold-start training.

**Reinforcement Learning Data** Our RL training builds on recent public datasets for LLM reasoning. Early efforts emphasize math (He et al., 2025; Yu et al., 2025; Luo et al., 2025b) and code (Luo et al., 2025a; Li, 2024; Mattern et al., 2025; Jain et al., 2024), while newer releases broaden coverage to STEM, logic, simulation, and tabular reasoning (Ma et al., 2025; Akter et al., 2025; Lin et al., 2025; Li et al., 2025; Cheng et al., 2025; Stojanovski et al., 2025). Two resources are especially useful: Cheng et al. (2025) consolidates multi-domain datasets from prior work, and Stojanovski et al. (2025) provides a library with 100+ data generators and verifiers. We draw primarily from these public releases and use the standard verifiable rewards they provide. Our training domains are summarized below.

- Math. Challenging problems from exams, practice sets, and competitions with verifiable final answers.
- STEM. Questions collected from QA sources and refined with LLMs. Subjects span physics, chemistry, business, history and more; answers may be numeric, symbolic expressions, or booleans.
- Code. Coding challenges from competitive programming and LeetCode-style datasets with unit tests.
- Simulation. Tasks adapted from code-based environments that require procedural simulation within the chain of thought, such as predicting program outputs (forward simulation) or inferring inputs for a given output (backward simulation).
- Logic. Datasets emphasizing constraint satisfaction and formal deduction.
- Tabular. Problems that require parsing, querying, and reasoning over one or more tables to synthesize the final answer.

### 3.2 TRAINING SETUP

We experiment with two models: Qwen3-4B (Yang et al., 2025) and Llama-3.1-8B (Grattafiori et al., 2024) since they strike a practical balance of model performance and training cost.

*Cold-Start SFT.* We use Axolotl (Axolotl, 2025) with AdamW (Loshchilov & Hutter, 2017). The peak learning rate is $5 \times 10^{-5}$ with 10% linear warmup, then decays to $0.1\times$ the peak. Training runs for 4 epochs. The same hyperparameters are used for both backbones.

*Reinforcement Learning.* We use `verl` (Sheng et al., 2024) with AdamW. The learning rate is $1 \times 10^{-6}$ with 10 warmup steps and then decays to 0. The prompt batch size is 256; for each prompt we sample 16 responses with temperature 1.0. The maximum input length is 4096 tokens and the maximum output length is 8192 tokens.

### 3.3 EVALUATION BENCHMARKS

We evaluate across six domains using widely adopted benchmarks: Math (AIME24; MATH500 (Hendrycks et al., 2021)), Code (HumanEval; MBPP; LiveCodeBench (Chen et al., 2021; Austin et al., 2021; Jain et al., 2024)), STEM (GPQA; SuperGPQA (Rein et al., 2023; Team et al., 2025)),

Logic (Zebra; Knights and Knaves; BoxNet (Lin et al., 2025; Stojanovski et al., 2025)), Simulation (CodeI/O; CRUXEval (Li et al., 2025; Gu et al., 2024)), and Tabular (HiTab; MultiHiertt; FinQA (Cheng et al., 2021; Zhao et al., 2022; Chen et al., 2022)).

### 3.4 BASELINES

We compare our models to several recent reasoning models that are trained with public data on math or general domains: (1) General Reasoner (Ma et al., 2025), (2) SimpleRL-Zoo (Zeng et al., 2025) (3) Guru (Cheng et al., 2025). In addition, we also compare reasoning curriculum to two variants where some components are removed: 1) cold start + joint RL where math-RL is removed, 2) direct joint RL where both cold-start and math-RL are removed. These comparisons would help us understand the contributions of each component in our curriculum.

## 4 RESULTS

### 4.1 RESULTS ON QWEN

The Qwen results are summarized in Table 1. Across all domains, the 4B model trained with reasoning curriculum (`RC-Qwen`) consistently outperforms similarly sized baselines: Guru-7B, General-Reasoner-7B, and SimpleRL-7B. Despite its smaller size, `RC-Qwen` is competitive with, and in several cases exceeds, 32B baselines. Relative to SimpleRL (trained primarily on math), `RC-Qwen` matches or surpasses it on math benchmarks and delivers clear gains on most non-math tasks. Compared with Guru-32B (trained on diverse domains and similar data as ours), `RC-Qwen` is competitive on the majority of tasks and leads on six benchmarks, supporting our claim that a math-first curriculum followed by joint cross-domain RL yields strong general reasoning in compact models.

### 4.2 RESULTS ON LLAMA

Table 2 reports results on Llama. Simply porting the Qwen recipe to Llama-3.1-8B yielded negligible gains, so we introduced two adjustments. First, we initialized from the instruct model (Llama-3.1-8B-Instruct)[1] rather than the base model, because the base model does not reliably follow instructions, which complicates reward extraction and impedes learning. Second, within the Math-RL stage we added a difficulty curriculum with two sub-stages: medium problems followed by hard problems. This curriculum made learning more stable and enabled a smooth handoff to joint RL. Because most prior work on RL for general reasoning evaluates Qwen models, directly comparable Llama baselines are scarce (Ma et al., 2025; Cheng et al., 2025; Hu et al., 2025a; Akter et al., 2025). Against our internal baselines, RL (direct joint RL) and CS+RL (cold start + joint RL), the curriculum consistently improves performance across all domains, supporting the claim that math-first elicitation followed by cross-domain RL is effective for Llama.

### 4.3 COGNITIVE SKILLS USAGE

We compare cognitive skill frequencies across models trained with Direct Joint RL (RL), Cold-Start + Joint RL (CS+RL), and our Reasoning Curriculum (`RC`). Following prior work (Gandhi et al., 2025; Zeng et al., 2025), we use GPT-4o-mini to tag four skills: subgoal setting, enumeration, backtracking, and verification. Figure 2 summarizes the results (upper: Qwen3-4B; lower: Llama-3.1-8B). Overall, `RC` increases the frequency of these skills for both backbones, supporting our hypothesis that math-first training improves cognitive skills across domains via the reasoning curriculum. Also, two observations are noteworthy. First, all settings exhibit a similarly high rate of subgoal setting (often near 100%), which suggests that it is necessary but not sufficient for solving complex problems. Second, CS+RL can show comparable rates of advanced skills in certain domains (for example, backtracking in Tabular for Qwen and verification and backtracking in Simulation for Llama). This suggests that Cold-Start helps models quickly imitate surface-level reasoning patterns, but on-policy training in the Math-RL stage appears important for fully consolidating the skills and converting them into the performance gains observed under the full `RC` pipeline.

---

[1] https://huggingface.co/meta-llama/Llama-3.1-8B-Instruct

Table 1: Evaluation Results on Qwen.

| | 32B | | 7B | | | 4B | | |
|---|---|---|---|---|---|---|---|---|
| **Task** | GURU | SimpleRL | GURU | General Reasoner | SimpleRL | RL | CS+RL | Reasoning Curriculum |
| *Math* | | | | | | | | |
| AIME-24 | 34.89 | 27.20 | 17.50 | 17.08 | 15.60 | 26.56 | 27.71 | **32.60** |
| Math-500 | 86.00 | 89.60 | 77.25 | 70.40 | 87.00 | 83.20 | 85.20 | **89.00** |
| *STEM* | | | | | | | | |
| GPQA | 50.63 | 46.46 | 40.78 | 38.64 | 35.98 | 45.83 | 48.99 | **53.16** |
| SuperGPQA | 43.60 | 37.73 | 31.80 | 30.64 | 27.29 | 33.00 | 39.60 | **41.40** |
| *Code* | | | | | | | | |
| HumanEval | 90.85 | 81.25 | 82.62 | 61.12 | 58.08 | 88.79 | 89.55 | **90.85** |
| LiveCodeBench | 29.30 | 19.80 | 16.49 | 8.51 | 6.72 | 23.66 | 23.21 | **26.34** |
| MBPP | 78.80 | 76.75 | 70.00 | 39.80 | 49.60 | 72.40 | 75.80 | **80.00** |
| *Simulation* | | | | | | | | |
| CodeIO | 12.63 | 9.75 | 15.63 | 7.13 | 6.63 | 6.13 | 14.75 | **20.63** |
| CruxEval-I | 80.63 | 72.63 | 61.72 | 63.63 | 56.25 | 70.75 | 78.13 | **82.13** |
| CruxEval-O | 88.75 | 67.75 | 71.28 | 56.50 | 58.31 | 71.50 | 76.25 | **79.75** |
| *Logic* | | | | | | | | |
| Knights Knaves | 17.62 | 16.22 | 14.43 | 14.73 | 15.26 | 65.94 | 68.69 | **71.10** |
| BoxNet | 0.12 | 0.25 | 1.06 | 1.60 | 0.78 | 83.85 | 88.77 | **93.80** |
| Zebra | 45.21 | 1.16 | 39.40 | 0.07 | 0.62 | 40.51 | 40.11 | **44.07** |
| *Tabular* | | | | | | | | |
| FinQA | 46.14 | 45.41 | 34.70 | 34.33 | 35.10 | 42.69 | 44.50 | **45.14** |
| HiTab | 82.00 | 69.00 | 74.20 | 54.40 | 50.40 | 73.80 | 71.30 | **76.60** |
| MultiHiertt | 55.28 | 52.83 | 44.94 | 31.62 | 37.57 | 52.38 | 50.30 | **54.02** |

RL = direct joint RL; CS+RL = cold-start then joint RL.

## 4.4 ABLATIONS

We ablate the components of the reasoning curriculum. Table 3 reports average performance. Removing the Math-RL stage, that is, using CS+RL (Cold-Start followed by Joint RL), reduces performance relative to the full curriculum. Removing Cold-Start as well, i.e., direct joint RL, leads to a further drop. The same pattern is observed for both Qwen and Llama models. These results indicate that each component contributes meaningfully to the performance of reasoning curriculum.

## 4.5 IMPROVEMENTS ACROSS REASONING CURRICULUM

We track performance across the curriculum stages (Cold-Start, Math-RL, and Joint-RL) in Figure 3 (top: Qwen3-4B; bottom: Llama-3.1-8B). In each sub-figure, the $y$-axis is the average score within a domain and the $x$-axis indexes the curriculum stage. Three patterns are consistent across both backbones. First, in Math, STEM, and Tabular, scores improve stage by stage: Math-RL exceeds Cold-Start, and Joint-RL further improves over Math-RL, suggesting shared reasoning representations across these domains. Second, in Simulation and Code, Math-RL reduces performance relative to Cold-Start even though both stages use only math data, indicating possible overfitting to math. Joint-RL however recovers the drop, and the full curriculum still outperforms the variant that skips Math-RL (see the CS+RL columns in Tables 1 and 2). Third, in Logic, performance is near zero after Cold-Start and Math-RL, implying that logic requires domain-specific training. Nevertheless, these stages appear to have a latent positive effect: under the full curriculum, logic accuracy surpasses direct joint RL (compare the RL column with Reasoning Curriculum in Tables 1 and 2).

Table 2: Evaluation Results on Llama.

| Task | RL | CS+RL | Reasoning Curriculum |
|---|---|---|---|
| *Math* | | | |
| AIME-24 | 7.40 | 9.58 | **14.37** |
| Math-500 | 55.60 | 69.60 | **74.40** |
| *STEM* | | | |
| GPQA | 32.94 | 35.86 | **39.90** |
| SuperGPQA | 27.50 | 29.50 | **31.70** |
| *Code* | | | |
| HumanEval | 70.27 | 69.82 | **74.24** |
| LiveCodeBench | 15.68 | 17.74 | **18.46** |
| MBPP | 60.40 | 58.80 | **64.00** |
| *Simulation* | | | |
| CodeIO | 10.75 | 16.25 | **17.38** |
| CruxEval-I | 50.50 | 61.00 | **65.50** |
| CruxEval-O | 23.00 | 61.00 | **60.62** |
| *Logic* | | | |
| Knights Knaves | 64.47 | 66.67 | **67.63** |
| BoxNet | 74.11 | 75.20 | **96.23** |
| Zebra | 35.43 | 32.86 | **41.08** |
| *Tabular* | | | |
| FinQA | 27.79 | 33.70 | **35.33** |
| HiTab | 74.90 | 75.30 | **78.30** |
| MultiHiertt | 40.25 | 38.99 | **44.05** |

RL = direct joint RL; CS+RL = cold-start then joint RL

Table 3: Ablations on training curriculum.

| Ablation | Qwen3-4B | Llama-3.1-8B |
|---|---|---|
| Reasoning Curriculum | **61.29** | **51.45** |
| − Math-RL | 57.68 | 46.99 |
| − Math-RL, − CS | 55.06 | 41.94 |

− Math-RL removes math RL; − CS further removes cold-start.

## 5 RELATED WORK

### 5.1 LLM REASONING

A key breakthrough in eliciting reasoning from LLMs is Chain-of-Thought (CoT) prompting (Wei et al., 2022), which asks models to produce intermediate steps before the final answer. Building on this foundation, recent proprietary models have pushed the boundaries of LLM reasoning by combining massive model scale with large-scale RL. OpenAI's GPT-o1 (OpenAI, 2024), for instance, leverages RL to explore and refine long, complex reasoning chains. This approach has demonstrated unprecedented performance on highly challenging domains like competitive math and programming.

The success of this paradigm has inspired the open-source efforts to develop similar capabilities. Models like QwQ (Team, 2024) and DeepSeek-R1 (Guo et al., 2025) take a similar RL approach and achieve results competitive with leading proprietary models. These efforts have also helped demystify the training process. Community ablations scrutinize when zero or minimal warm-up succeeds and how base model choice affects outcomes (Zeng et al., 2025). There is also evidence that careful scaling and length control can push small models to strong results, for example DeepScaleR-1.5B and DeepCoder-14B, which report competitive performance on verifiable benchmarks (Luo et al.,

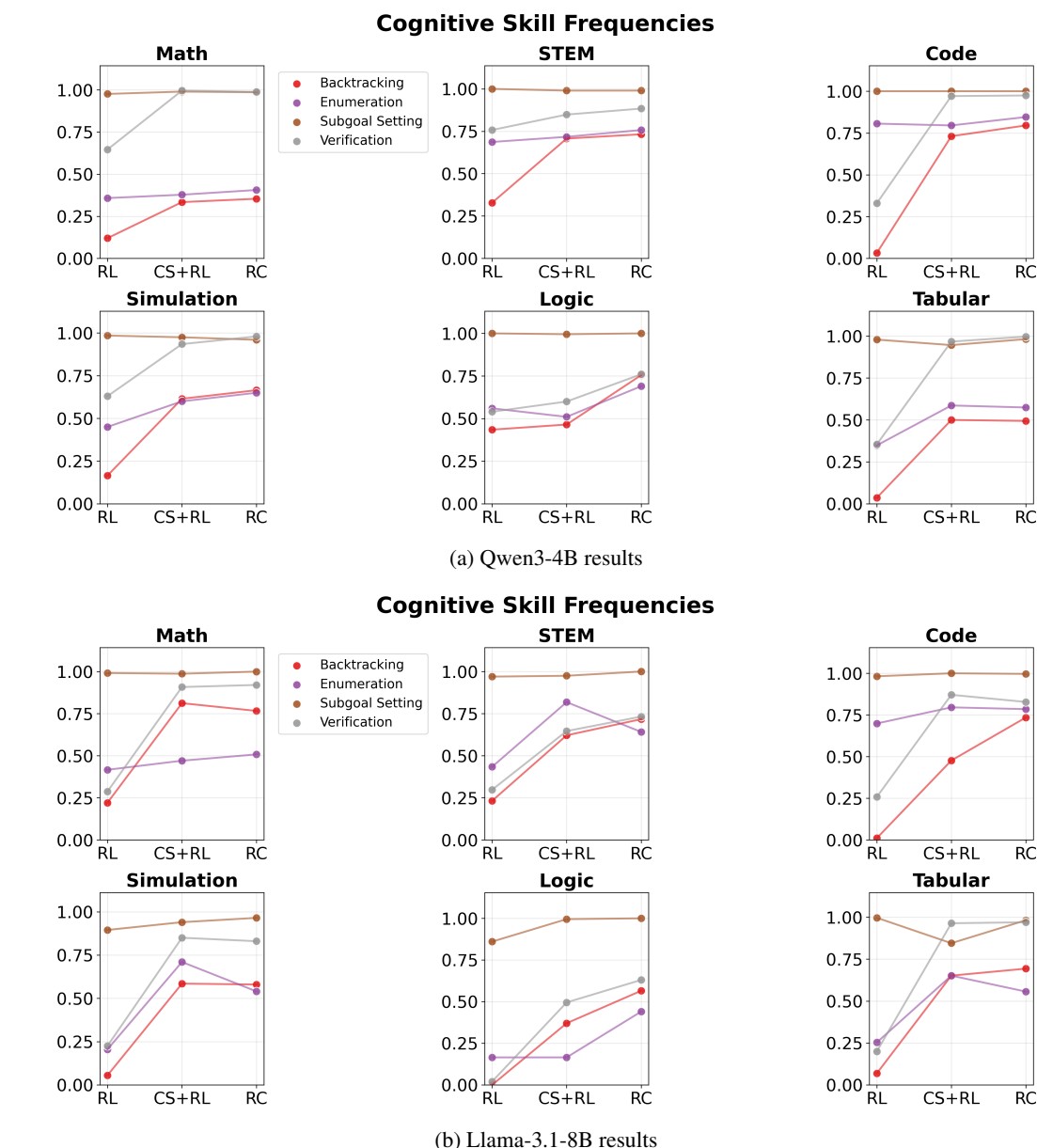

(a) Qwen3-4B results

(b) Llama-3.1-8B results

Figure 2: Cognitive skill frequencies by training setting. RL = direct joint RL; CS+RL = cold-start then joint RL; RC = reasoning curriculum. Top: Qwen3-4B; bottom: Llama-3.1-8B.

2025b;a). Intriguingly, recent studies show that substantial gains on math can be triggered by weak or even misleading reward signals, including rewards that are random or known to be incorrect (Shao et al., 2025b), and in extreme cases by training on a single example (Wang et al., 2025). This sensitivity of math reasoning to RL supervision motivates our approach: we leverage these dynamics to improve reasoning across domains through a cross-domain reasoning curriculum.

## 5.2 REASONING ACROSS DOMAINS

Despite rapid progress, most open research concentrates on math and code, where a large amount of data is available and rewards are easily verifiable. Recent efforts have begun to expand coverage beyond these areas. Akter et al. (2025) and Ma et al. (2025) curate STEM datasets with verifiable rewards, exploiting the ease of multiple-choice verification and using LLMs to normalize and com-

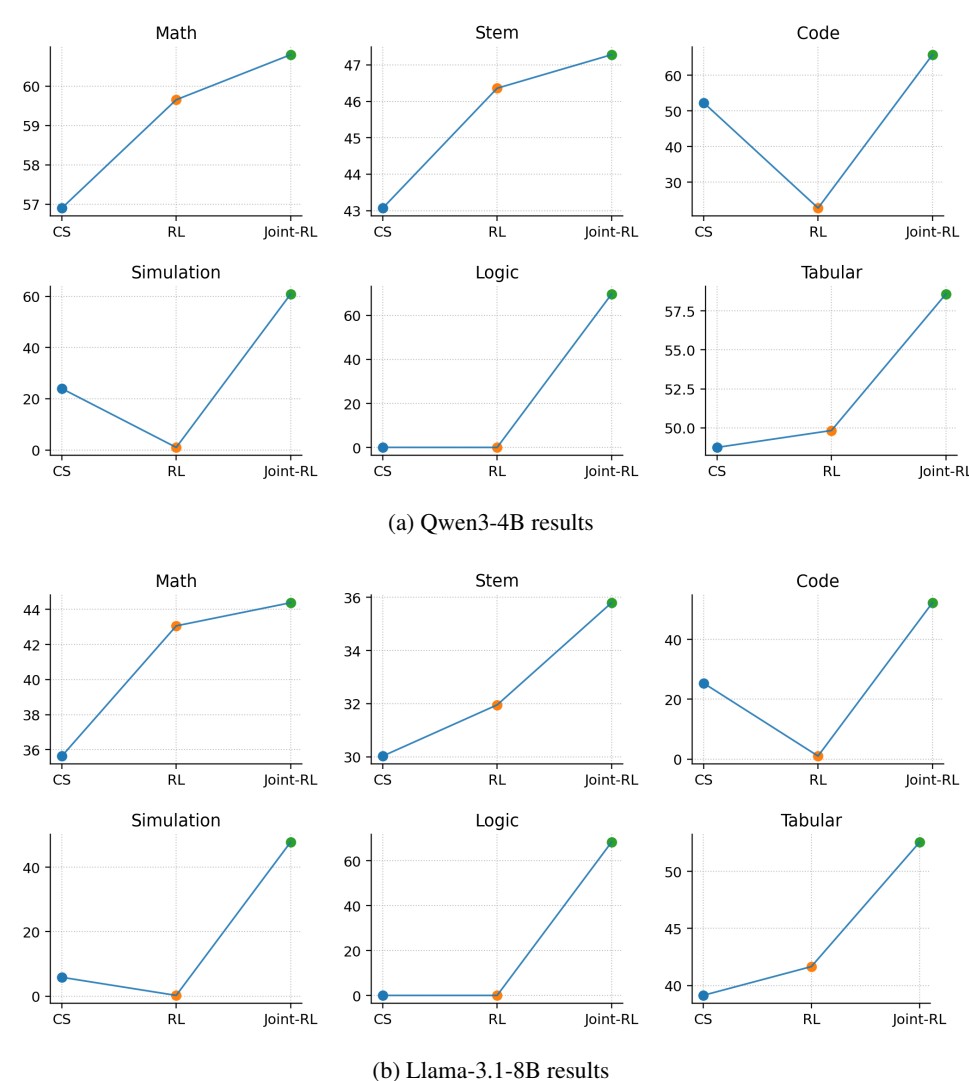

(a) Qwen3-4B results

(b) Llama-3.1-8B results

Figure 3: Trends across curriculum stages by task. CS = Cold-Start; RL = Math-RL; Joint-RL = RL on mixed-domain data. Top: Qwen3-4B; bottom: Llama-3.1-8B. Each point shows the average score within a domain at each stage.

pare answers across varied surface forms. Building on such resources, Cheng et al. (2025) introduce Guru, which further incorporates logic, simulation, and tabular domains. Collectively, these works advance data collection, cleaning, and cross-domain evaluation, revealing distinct performance patterns across tasks. In our work, we leverage these multi-domain resources and other logic datasets to study how to train a strong reasoning model across domains.

# 6  CONCLUSION

We introduced `Reasoning Curriculum`, a minimal two-stage curriculum that first elicits reasoning skills in math through cold start and RL, then adapts and refines them with joint RL across diverse domains. On Qwen3-4B and Llama-3.1-8B, `Reasoning Curriculum` delivers consistent multi-domain gains. Ablations show that both stages are necessary, and a cognitive-skill analysis indicates increased use of advanced behaviors such as verification and backtracking. The recipe is backbone-agnostic and relies only on standard verifiability checks, which makes it easy to adopt.

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

## A  APPENDIX

### THE USE OF LARGE LANGUAGE MODELS (LLMS)

We used large language model (LLM) assistants to improve the clarity of the manuscript. Allowed uses included: suggesting word choices, fixing grammatical errors, and smoothing sentences and transitions. All generated edits were reviewed and, when necessary, rewritten by the authors.

