# OpenReview forum: "Reasoning Curriculum: Bootstrapping Broad LLM Reasoning from Math"
_ICLR.cc/2026/Conference — ICLR 2026 Conference Withdrawn Submission_

### Official Review · Reviewer_DMo9 · 2025-10-23

**Soundness:** 1
**Presentation:** 3
**Contribution:** 2
**Rating:** 2
**Confidence:** 4

**Summary:**

This work is focused on the RL reasoning stage for LLMs. The authors propose to first perform RL on math-only problems, and later introduce a wider range of domains. The authors validate this two-stage procedure, which they call "Reasoning Curriculum," with two base models from the Qwen and Llama families.

**Strengths:**

- The proposed method is intuitive, and the motivation is well-introduced. As math is an amenable domain for RL, it makes sense that priming the model with these skills facilitates future broader reasoning.
- Overall, the paper is well-written and free of typos. The methodology and the past techniques used are clearly presented.
- The domain of RL reasoning is a relevant topic for language model research and the ICLR community.

**Weaknesses:**

**Main**
1. The choice of baselines seems very questionable to me, and the statements like the method "sometimes exceeds, 32B systems" are never properly contextualized: the model is only compared with baselines from the older Qwen2.5 model family, which is significantly weaker than the Qwen3 model family. Once again, the authors use Qwen3-4B as their Qwen model choice, not Qwen2. Yet, results for even baselines comparing the authors' model to just the performance of plain Qwen 3 4B, Qwen3 7B, and Qwen3 32B **instruct** versions are never even provided. For statements such as the one made in the introduction referenced above, the method would have to outperform at least these models. I think the odd baseline choice is particularly evident for "general reasoner": the authors of the general reasoner paper even provide both a 4B model with Qwen3 and a 7B model with Qwen2.5, which is significantly less performant (over 50% worse on some tasks from the general reasoner paper). Yet, this paper chooses to evaluate with *only* the older, weaker, and larger 7B model for a more favourable comparison. I hope I am not missing something, but this does not seem to ever be explicitly mentioned/motivated in the paper, and I had to go back to the general reasoner paper to find out.
2. While data-ordering is the key methodological contribution, I think an ablation doing cold-start data and later training on both math data and joint RL data together at the same time should have been provided. It's unclear to me why the authors instead report a joint RL ablation "where math-RL is removed" [225] with and without SFT, introducing many confounding factors, as at the moment, the authors do not even seem to evaluate their own reasoning curriculum without the cold start phase.
3. While the authors characterize their cold start phase as "brief" (line 123), it actually involved training for four epochs on tens of thousands of samples. DeepSeek R1 is a model known to provide highly effective distillation data. Yet, results with Qwen and Llama with and without the SFT procedure (and no other RL phase) are not provided. This makes it hard to put into perspective the contribution of the proposed pipeline.
4. No experimental details or code are provided (the paper does not even have an Appendix Section). Given that the proposed method involves several stages of distillation using DeepSeek R1 data and later RL, the results seem hardly reproducible.
5. In its implementation, the curriculum is considerably modified for the llama model, the same curriculum. This puts into question the generality of the proposed method as a general recipe for models beyond Qwen, and goes in stark contrast with statements such as "The recipe is backbone-agnostic and relies only on standard verifiability checks, which makes it easy to adopt." [485]


**Other**
1. The proposed pipeline is very expensive and slow to train.  First, a large amount of synthetic data must be generated by querying DeepSeek R1 on 20K questions. Then, an LLM is used to compute rewards for STEM questions. Yet, this potential limitation is never explicitly.
2. The fact that LLMs' post-training behavior is due to their pre-training and potential data contamination is not a new hypothesis (e.g. [1], [2]). Yet, this presented as novel in the text:e.g., "Given the readiness of skill elicitation in the math domain, we hypothesize that pretraining already exposes models to these skills in data-rich domains such as math, making them easier to elicit during post-training" [105-107]..

[1] Shao, Rulin, et al. "Spurious rewards: Rethinking training signals in rlvr." arXiv preprint arXiv:2506.10947 (2025).

[2] Wu, Mingqi, et al. "Reasoning or memorization? unreliable results of reinforcement learning due to data contamination." arXiv preprint arXiv:2507.10532 (2025).

**Questions:**

I have raised my main areas of concern in the Weaknesses Section above, such as the questionable evaluation with many confounding factors and lack of reproducibility. I would encourage the authors to address these aspects in future revisions.

---

### Official Review · Reviewer_2n1V · 2025-10-31

**Soundness:** 3
**Presentation:** 2
**Contribution:** 2
**Rating:** 4
**Confidence:** 3

**Summary:**

The paper proposes Reasoning Curriculum, a minimal two-stage reinforcement learning pipeline to elicit and generalize reasoning capabilities in large language models.
Stage 1: A brief supervised cold start on math problems, followed by math-only RL using verifiable correctness-based rewards to elicit cognitive skills such as verification and backtracking.
Stage 2: Joint RL across multiple domains (math, code, STEM, logic, simulation, tabular) to transfer and consolidate these skills. Experiments on Qwen3-4B and Llama-3.1-8B demonstrate consistent gains across domains.

**Strengths:**

1. The recipe is simple and practical;
2. Strong and Broad Empirical Results across domains;
3. Works on both Qwen and Llama;

**Weaknesses:**

1. Limited Comparison with Strong Baselines;
2. Missing head-to-head evaluations against recent strong open-source RL pipelines.
3. Benchmarks are mostly verifiable tasks; free-form reasoning is not evaluated.

**Questions:**

Same to Weakness.

---

### Official Review · Reviewer_LPbY · 2025-11-03

**Soundness:** 1
**Presentation:** 2
**Contribution:** 1
**Rating:** 2
**Confidence:** 3

**Summary:**

The authors propose a two-stage curriculum-based approach for eliciting stronger reasoning abilities in LLMs. They employ various ablations to demonstrate that their curriculum improves accuracy across a variety of reasoning domains, and claim that it performs competitively with comparable attempts.

**Strengths:**

The authors study a timely and interesting topic. The prose is well-written, though vague and doesn't seem to contextualize its results well against similar work (see Weaknesses below).

**Weaknesses:**

The authors are vague on the details of their evaluation and comparison with similar projects (GURU, SimpleRL, etc.).
* It's unclear how they ran their evaluations. What inference settings did they use specifically?
* Did they re-run baselines from other projects using their own setup, or copied numbers from the other papers?
* If they re-ran baselines, how did they do so exactly? Did they use their own grading/inference rig or standardize evaluations in some way across all baseline models?

There also seems to be a strong possibility of training data contamination. For instance, in Table 1, performance in BoxNet seems to jump from near 0 for many of the external models to almost perfect for the authors' model. Looking at Section 3, it appears that BoxNet was cited under both the *Reinforcement Learning Data* section and the *Evaluation Benchmark* section. This seems to imply that the authors may have included BoxNet in the training diet? Were the authors careful to separate train from test examples? If tasks like BoxNet require specialized input formats that require finetuning, how were other baselines evaluated fairly?

In general, the description of training and evaluation data was vague, and does not seem to specify how test examples are kept disjoint from training examples. Where did train and test examples come from specifically? Which specific datasets did they use, and how? Section 3.1 only seems to give a broad gloss over dataset categories, but provides no specifics on exact datasets employed and how they were prepared. Given some of the unusually high numbers in Table 1 paired with the missing details, it is difficult to judge the significance of their results, and whether they are conflated by data contamination.

Finally, the model sizes explored in the paper seem to be fairly small. Benefits from RL refinements may diminish or disappear at larger scales. Were the authors able to test whether benefits from their method continue to hold for larger models?

**Questions:**

See Weaknesses above

---

### Official Review · Reviewer_VPgQ · 2025-11-04

**Soundness:** 1
**Presentation:** 3
**Contribution:** 2
**Rating:** 2
**Confidence:** 4

**Summary:**

The paper introduces a two-stage reasoning curriculum for RL that uses math as a central training task to unlock general reasoning in small open models (4B, 8B). There are two stages in the proposed method. Stage 1 does cold start SFT and do RL only on math data, where rewards are easy to verify, which strongly elicits core skills like backtracking and verification as suggested by the authors. Stage 2 then does joint RL over math, STEM, code, simulation, logic and tabular tasks, which tries to transfer those skills across domains. Experiments on Qwen3-4B and Llama3.1-8B shows gains across many benchmarks. Ablations show that both the math-first RL stage and the subsequent mixed-domain RL are necessary. Analysis of generated solutions confirms that the curriculum increases the use of more sophisticated reasoning strategies rather than just surface-level pattern matching.

**Strengths:**

* The paper tries to extend the capability learned from math tasks to other domains, which is a worth studying topics to enhance the reasoning in LLMs. The paper uses four kinds of skills to track the cognitive capability that assist reasoning, and they also provide quantitive results on this.
* Experiments show that some of the coginitive patterns can be transferred from math tasks to other domain tasks like STEM and coding, using the proposed reasoning curriculum method. Benchmarking on different task domains also support the claims.
* The paper is well-written and easy to follow. The method presentation is clear.

**Weaknesses:**

* The conclusion from this paper is not new. The method lacks significant contribution as a ICLR submission.  The so called "curriculum" seems to be simply math (SFT + RL) then all-domain data. There are already a few discussion regarding math reasoning capability can be transferred to other or more general domains, e.g., [1,2].
* Unfair comparison. Guru uses Qwen2.5 series models (other baselines in Table 1 also use inferior base models compared with authors') as the backbone and this paper uses Qwen3 series. Qwen3 series is reported to be trained on larger data. Also, Qwen3 models often matches or beats the next size up in Qwen2.5 series. For example, Qwen3-4B may already match or beats the performance of Qwen2.5-7B. In Table 1, the authors compare their 4B-Reasoning-Curriculum model with other baselines (e.g., Guru, SimpleRL) and claiming that their method is superior, which is problematic. The authors should at least use the same backbone model (Qwen2.5 series) to compare with them. Or the authors can re-train the baselines with Qwen3 models. The unfair comparison in Table 1 significantly ruins the soundness of the paper, leaving doubt whether the proposed method is effective.




References

[1] Huan, Maggie, et al. "Does Math Reasoning Improve General LLM Capabilities? Understanding Transferability of LLM Reasoning." arXiv preprint arXiv:2507.00432 (2025).

[2] Akter, Syeda Nahida, et al. "Nemotron-crossthink: Scaling self-learning beyond math reasoning." arXiv preprint arXiv:2504.13941 (2025).

**Questions:**

* For the ablation experiments, did you make sure that the optimization steps for each method is the same? Specifically, do "RL", "CS+RL" and "Reasoning Curriculum" have the same optimization steps?

---

### Note · Authors · 2025-12-18

I have read and agree with the venue's withdrawal policy on behalf of myself and my co-authors.